# Sustainable Optimal Control for Switched Pollution-Control Problem with Random Duration

**DOI:** 10.3390/e25101426

**Published:** 2023-10-08

**Authors:** Yilun Wu, Anna Tur, Hongbo Wang

**Affiliations:** 1College of Electronic Science and Engineering, Jilin University, Jilin 130012, China; wuyilun310@gmail.com; 2Faculty of Applied Mathematics and Control Processes, Saint Petersburg University, Saint Petersburg 199034, Russia; a.tur@spbu.ru

**Keywords:** differential game, cooperative game, switched system, random duration, sustainability

## Abstract

Considering the uncertainty of game duration and periodic seasonal fluctuation, an *n*-player switched pollution-control differential game is modeled to investigate a sustainable and adaptive strategy for players. Based on the randomness of game duration, two scenarios are considered in this study. In the first case, the game duration is a random variable, $T_{f}$, described by the shifted exponential distribution. In the second case, we assumed that players’ equipment is heterogeneous, and the *i*-th player’s equipment failure time, $T_{fi}$, is described according to the shifted exponential distribution. The game continues until a player’s equipment breaks down. Thus, the game duration is defined as $T_{f}=min\left\{T_{f1},\ldots ,T_{fn}\right\}$. To achieve the goal of sustainable development, an environmentally sustainable strategy and its corresponding condition are defined. By using Pontryagin’s maximum principle, a unique control solution is obtained in the form of a hybrid limit cycle, the state variable converges to a stable hybrid limit cycle, and the total payoff of all players increases and then converges. The results indicate that the environmentally sustainable strategy in the *n*-player pollution-control cooperative differential game with switches and random duration is a unique strategy that not only ensures profit growth but also considers environmental protection.

## 1. Introduction

Practical ecological, economic, and engineering problems comprise switching phenomena [1,2,3]. All systems involving logical decision-making and continuous (smooth) dynamics, such as robot systems [4], chemical control processes [5], etc., can be transformed into hybrid systems with multiple regimes of dynamics.

Therefore, the hybrid dynamic system has gained considerable a research interest in the environmental, economic, and engineering fields. In addition, changed systems, in the form of time-driven and state-driven switches, are increasingly common. The up-to-date contributions to those fields are [6,7,8,9,10]. In [9,11], an optimal solution in the form of a hybrid limit cycle (HLC) was introduced as the best possible candidate for the infinite-horizon optimization problem. However, the results were only about the optimal control of a single agent and did not explore the optimal control strategy of multiple players.

In addition, in contrast to the general pollution-control problems with deterministic terminal time [12,13] or with infinite horizon [10,14], the randomness of the game duration cannot be ignored, because the game may end abruptly. The reasons behind this can be an equipment break-down, an economic failure. or a natural disaster, among many others [15]. In [15,16], differential pollution-control games with random duration were thoroughly analyzed. However, the impact of seasonal fluctuation on the system was not considered.

Thus, by combining the two aforementioned research directions, an *n*-player cooperative differential game was explored for pollution control. The game involves infinite time-driven switches and encompasses random game duration.

The contributions of this paper are summarized as follows:A novel model is proposed to address challenges within the context of an *n*-player cooperative differential game for pollution control with time-driven switches and random duration. The time-driven switches within system are denoted by a periodic piecewise-constant function. Taking into account the randomness of game duration and the players’ equipment warranty period, the finite-horizon optimal control problem is reformulated as an infinite-horizon optimal control problem, in which the game duration is modeled considering two scenarios based on shifted exponential distributions. The proposed model introduces innovative concepts and refines previously established methodologies, aiming to enhance its adaptability to real-world scenarios and yield more practical outcomes.In addition, this study proves the sustainability and uniqueness of the environmentally sustainable solution upon the proposed model.Solving the optimal pollution-control problem with time-driven switches and random duration: We employed Pontryagin’s maximum principle and thoroughly analyzed the adjoint variable dynamics to derive the shifted equilibrium value, resulting in a unique environmentally sustainable solution in the form of an HLC, which is environmentally friendly and guarantees the profit of all players.

The obtained results can be applied to the optimization problems of switched systems with periodic switching signals, such as the formation control problem [17] and capital investment problem [8]. Additionally, they can provide valuable insights on how agents can effectively adapt to a rapidly changing and evolving environment, allowing them to reap the benefits.

The remainder of this paper is structured as follows. In Section 2, the model of the *n*-player pollution control differential game with time-driven switches and random duration is formulated. In addition, considering different types of game durations, two case studies were conducted. Section 3 presents a discussion of the cooperative game with identical shifted exponential distribution and provides a the definition of the environmently sustainable control and determines the equilibrium value of adjoint variable that is used to derive the unique solution. In Section 4, we discuss a cooperative game with different shifted exponential distributions and obtain the corresponding equilibrium value and unique solution. Section 5 provides an illustration of a pollution-control differential game involving two players, and optimal solutions for both scenarios are demonstrated numerically.

## 2. The Problem Statement

All notations used through the paper are summarized in Appendix A, Table A1.

In this study, we consider the optimal lake-pollution-control game model based on [15,16,18]. The game involves *n* players (factories). Each player *i* manages his/her emissions policy toward the lake, such that the dynamic of the pollution stock is governed by the linear differential equation with the following initial condition:(1)$$\dot{z}=\sum_{i=1}^{n}\xi_{i}v_{i}-\delta \left(t\right)z,\;z\left(0\right)=z_{0},$$
where *z* is the stock of pollution within a fixed natural reservoir (e.g., a lake), $v_{i}\in \left[0,b_{i}\right],i\;=\;\bar{1,n}$ denotes the emissions rate, $b_{i}$ is the maximal admissible emissions rate of each player, $\xi_{i}\in \left(0,1\right)$ is the fraction of the emitted pollutants accumulated in the reservoir from each player (e.g., factory), and δ(t) is the self-cleaning rate of the reservoir. Furthermore, we have $z_{0}\geq 0$ and the state z(t) is non-negative for all t≥0.

The self-cleaning rate for lakes is widely acknowledged to vary throughout the year. This could be due to the impact of various factors, including temperature and light fluctuations, during specific periods (e.g., over the span of a year). Considering the impact of external seasonal changes on the lake, it is reasonable to postulate that the self-cleaning rate of the lake is not a fixed constant; however, it varies as a function of time. Thus, we further assumed that the self-cleaning rate of reservoir δ(t) is represented by a periodic piecewise-constant function, which defined as the following mathematical expression:(2)$$\delta \left(t\right):=\left\{\begin{array}{cc} \delta_{1}>0, & t\in [kT,(k+\tau )T],\\ \delta_{2}>0, & t\in [(k+\tau )T,(k+1)T]. \end{array}\right.$$

The whole time duration $D=[0,T_{f}]$ is divided into equal periods of length *T*, each of which is subdivided into two parts: [kT,(k+τ)T] and [(k+τ)T,(k+1)T], where τ∈(0,1) is the switching ratio and $k\in N_{0}$. When the system is in the first subperiod, $\delta \left(t\right)=\delta_{1}$, whereas in the second subperiod, $\delta \left(t\right)=\delta_{2}$ and $\delta_{1}\neq \delta_{2}$.

Note that the production is usually assumed to be linearly related to the emissionss. Therefore, the revenue function can be expressed in terms of emissionss [18]. The revenue function of each player $R_{i}\left(v_{i}\right)$ is strictly concave. The marginal revenue decreases with the increasing emissions rate of each player $v_{i}\left(t\right)\in \left[0,b_{i}\right]$. However, zero emissions (production) is unprofitable. Each player incurs a damage cost, $C_{i}\left(z\right)$, for mitigating their emissions at moment *t*, and this cost function is increasing and convex. Thus, a quadratic revenue functional $R_{i}\left(v_{i}\right)=a_{i}v_{i}\left(b_{i}-v_{i}/2\right)$ [19] and a linear cost functional form $C_{i}\left(z\right)=q_{i}z$ are commonly derived to represent the instantaneous payoff of player *i* as $L\left(v_{i},z\right)\;=\;R_{i}\left(v_{i}\right)\;-\;C_{i}\left(z\right)$.

Then, the general form of the integral payoff for player *i* is as follows: (3)$$J_{i}\left(z_{0},v_{1},v_{2},\ldots ,v_{n}\right)=\int_{0}^{T_{f}}\left[a_{i}v_{i}\left(b_{i}-v_{i}/2\right)-q_{i}z\right]\;dt,$$
where $a_{i}>0$ is a positive constant used to transform the emissions flow to the profit flow. Coefficient $q_{i}>0$ is a positive constant, corresponding to the tax that a player must bear (e.g., an ecotax).

Considering the randomness of the game duration, we assumed that the terminal time of game $T_{f}$ is a random variable. After simplifying the integral payoff [20], the expected integral payoff of player *i* is obtained as follows:(4)$$\begin{array}{cc} J_{i}\left(z_{0},v_{1},v_{2},\ldots ,v_{n}\right) & =E(\int_{0}^{T_{f}}\left[a_{i}v_{i}\left(b_{i}-v_{i}/2\right)-q_{i}z\right]\;dt)\\ \; & =\int_{0}^{\infty }\int_{0}^{s}\left[a_{i}v_{i}\left(b_{i}-v_{i}/2\right)-q_{i}z\right]\;dt\;dF\left(s\right)\\ \; & =\int_{0}^{\infty }\left(1-F\left(t\right)\right)\left[a_{i}v_{i}\left(b_{i}-v_{i}/2\right)-q_{i}z\right]\;dt, \end{array}$$
where F(t) is the cumulative distribution function of $T_{f}$.

## 3. Cooperative Game with Identical Shifted Exponential Distribution

If all players share common pollution-control equipment (e.g., filters), we assume that the game duration is a realization of the random variable $T_{f}$ (the time of equipment failure is same for all players and coincides with the end of the game).

Assume that at the beginning of the game, all players use a new pollution-control equipment, and this equipment comes with a warranty period. The warranty period refers to a specified time frame after the sale of a product or service, during which the manufacturer or supplier assures free repairs or replacements. During this period, if the product experiences any quality issues or malfunctions, the consumer can avail free repairs/replacement services. Hence, before the warranty period, there is no risk of equipment breakdown, while after the warranty period, the equipment is subject to the risk of potential damage. The game terminates when the equipment breaks down.

Therefore, we consider a cooperative game for pollution control, wherein the randomness of game duration and consideration of the warranty period are represented by a shifted exponential distribution.

For the mathematical definition of random variable $T_{f}$, we applied a shifted exponential distribution, which is given by
(5)$$F\left(t\right)=\left\{\begin{array}{cc} 1-e^{\lambda (\theta -t)}, & t>\theta ,\\ 0, & t\in [0,\theta ], \end{array}\right.$$
where θ>0 is the shift parameter of the exponential distribution from the initiation of game 0, which represents the equipment warranty period. The equipment does not encounter the risk of failure before the warranty period, and after this period, θ, the failure rate is constant over time. In addition, λ>0 is the parameter of distribution and $E\left(T_{f}\right)=\theta +\frac{1}{\lambda }$.

By substituting (5) into (4), the whole time horizon is split into [0,+∞] as [0,θ]∪(θ,∞], according to Bellman’s optimality principle, and the overall payoff functional, (4), can be rewritten as a sum:(6)$$\begin{array}{cc} J_{i}\left(z_{0},v_{1},v_{2},\ldots ,v_{n}\right)= & \int_{0}^{\theta }\left[a_{i}v_{i}\left(b_{i}-v_{i}/2\right)-q_{i}z\right]\;dt\\ \; & +\int_{\theta }^{\infty }e^{\lambda (\theta -t)}\left[a_{i}v_{i}\left(b_{i}-v_{i}/2\right)-q_{i}z\right]\;dt. \end{array}$$

All players (factories) act together to maximize their joint payoff:(7)$$\begin{array}{cc} J_{co}\left(z_{0},v_{1},v_{2},\ldots ,v_{n}\right) & =\sum_{i=1}^{n}J_{i}\left(z_{0},v_{1},v_{2},\ldots ,v_{n}\right)\\ \; & =J_{co1}\left(z_{0},v_{1},v_{2},\ldots ,v_{n}\right)+J_{co2}\left(z_{\theta },v_{1},v_{2},\ldots ,v_{n}\right)\\ \; & =\sum_{i=1}^{n}\int_{0}^{\theta }\left[a_{i}v_{i}\left(b_{i}-v_{i}/2\right)-q_{i}z\right]\;dt\;+\\ \; & \sum_{i=1}^{n}\int_{\theta }^{\infty }e^{\lambda (\theta -t)}\left[a_{i}v_{i}\left(b_{i}-v_{i}/2\right)-q_{i}z\right]\;dt. \end{array}$$

By using Pontryagin’s maximum principle, the cooperative solution is obtained as a result of the joint optimization problem. We solved this optimization problem separately. First, the second interval (θ,∞] is considered with respect to players’ joint payoff $J_{co2}\left(z_{\theta },v_{1},v_{2},\ldots ,v_{n}\right)=\sum_{i=1}^{n}\int_{\theta }^{\infty }e^{\lambda (\theta -t)}\left[a_{i}v_{i}\left(b_{i}-v_{i}/2\right)-q_{i}z\right]\;dt$, and then the first interval [0,θ] is considered with respect to players’ joint payoff $J_{co1}=\sum_{i=1}^{n}\int_{0}^{\theta }\left[a_{i}v_{i}\left(b_{i}-v_{i}/2\right)-q_{i}z\right]\;dt$.

### 3.1. Second Interval—(θ,∞]

The payoff functional of the second interval (θ,∞] is
(8)$$J_{co2}\left(z_{0},v_{1},v_{2},\ldots ,v_{n}\right)=\sum_{i=1}^{n}\int_{\theta }^{\infty }e^{\lambda (\theta -t)}\left[a_{i}v_{i}\left(b_{i}-v_{i}/2\right)-q_{i}z\right]\;dt.$$

Further, the Hamiltonian is given by
(9)$$H_{co2}\left(t\right)=e^{\lambda (\theta -t)}\sum_{i=1}^{n}\left[a_{i}v_{i}\left(b_{i}-v_{i}/2\right)-q_{i}z\right]+\psi_{2}\left(\sum_{i=1}^{n}\xi_{i}v_{i}-\delta \left(t\right)z\right).$$

Then, we have the canonical system:(10)$$\begin{array}{cc} \; & \dot{z}=\sum_{i=1}^{n}\xi_{i}v_{i}-\delta \left(t\right)z,\\ \; & {\dot{\psi }}_{2}=e^{\lambda (\theta -t)}\sum_{i=1}^{n}q_{i}+\delta \left(t\right)\psi_{2}. \end{array}$$

Let ${\tilde{\psi }}_{2}=e^{-\lambda (\theta -t)}\psi_{2}$. Then, we have
(11)$${\dot{\tilde{\psi }}}_{2}=\sum_{i=1}^{n}q_{i}+\left(\delta \left(t\right)+\lambda \right){\tilde{\psi }}_{2}.$$

Based on the first order derivative of the Hamiltonian, the optimal control is obtained as follows:(12)$${\tilde{v}}_{i}^{*}=\left\{\begin{array}{cc} b_{i}, & {\tilde{\psi }}_{2}>0,\\ b_{i}+\frac{\xi_{i}}{a_{i}}{\tilde{\psi }}_{2}, & {\tilde{\psi }}_{2}\in \left[-\frac{a_{i}b_{i}}{\xi_{i}},0\right]\\ 0, & {\tilde{\psi }}_{2}<-\frac{a_{i}b_{i}}{\xi_{i}}. \end{array}\right.$$

For the payoff functional (8), which is defined in infinite horizon (θ,∞] with infinite time-driven switches, players should consider a sustainable development. Therefore, this study aimed to find a sustainable decision-making pattern so that players can find an optimal compromise between profit and penalty (e.g., ecotax).

**Definition 1.** 
*The optimal control, $v_{i}^{*}$, is environmentally sustainable if it does not take on boundary values, except at isolated instances of time, i.e., ${\tilde{\psi }}_{2}\in \left[-\frac{a_{i}b_{i}}{\xi_{i}},0\right]$, for all t≥θ.*


This definition is based on the long-term economic interests of all players. For $v_{i}^{*}>b_{i}$, the control (the rate of emissions) of player *i* remains at its maximum value. Evidently, this is not profitable because player *i* bears high ecotax. In another situation $v_{i}^{*}<b_{i}$, the revenue of player *i* cannot exceed the cost; therefore, the player must halt the production after a certain time interval to allow the stock of pollution to decrease to a lower level. Hence, both of these situations are not acceptable for sustainable production [21].

The following theorem and its proof are close to the result presented in [11]. The main difference is that the condition for the adjoint variable was not set at the initial moment but at moment θ. In addition, this value depends on the interval to which θ belongs.

**Theorem 1.** 
*The solution to (10) satisfying $z\left(0\right)=z_{0}$ and $\psi_{2}\left(\theta \right)=\psi_{hlc}$ with*

(13)
$$\psi_{hlc}=\left\{\begin{array}{cc} L\left(1-e^{p_{2}T\left(\tau -1\right)}\right)e^{p_{1}m}-\frac{q}{p_{1}}, & m\in [0,\tau T],\\ L\left(1-e^{p_{1}\tau T}\right)e^{p_{2}\left(m-T\right)}-\frac{q}{p_{2}}, & m\in [\tau T,T] \end{array}\right.$$

*is the unique optimal solution to (1)–(7). Here, $q=\sum_{i=1}^{n}q_{i},\;p_{1}=\delta_{1}+\lambda ,\;p_{2}=\delta_{2}+\lambda ,\;m=\theta -\left\lfloor \frac{\theta }{T}\right\rfloor T,\;L=\frac{q(p_{1}-p_{2})}{p_{1}p_{2}\left(e^{p_{2}T\left(\tau -1\right)}-e^{p_{1}\tau T}\right)}.$*


**Proof.** To obtain a periodic solution, the following equation is first solved:
(14)$${\tilde{\psi }}_{2}\left(\theta \right)={\tilde{\psi }}_{2}\left(\theta +T\right).$$Let $k_{1}=\left\lfloor \frac{\theta }{T}\right\rfloor$; then, $\theta \in [k_{1}T,\left(k_{1}+1\right)T]$. On interval $\left[k_{1}T,\left(k_{1}+1\right)T\right]$, the solution to (11) has the form
$${\tilde{\psi }}_{2}\left(t\right)=\left\{\begin{array}{cc} c_{1}e^{p_{1}t}-\frac{q}{p_{1}}, & t\in [k_{1}T,k_{1}T+\tau T],\\ c_{2}e^{p_{2}t}-\frac{q}{p_{2}}, & t\in [k_{1}T+\tau T,\left(k_{1}+1\right)T]. \end{array}\right.$$If $\theta \in [k_{1}T,k_{1}T+\tau T]$ or m∈[0,τT], then ${\tilde{\psi }}_{2}\left(\theta \right)=c_{1}e^{p_{1}\theta }-\frac{q}{p_{1}}$, which is the same thing, solving (14), we have
$${\tilde{\psi }}_{2}\left(\theta \right)=L\left(1-e^{p_{2}T\left(\tau -1\right)}\right)e^{p_{1}m}-\frac{q}{p_{1}}.$$If $\theta \in [k_{1}T+\tau T,\left(k_{1}+1\right)T]$, or, which is the same thing, m∈[τT,T], then ${\tilde{\psi }}_{2}\left(\theta \right)=c_{2}e^{p_{1}\theta }-\frac{q}{p_{2}}$. Solving (14) we have
$${\tilde{\psi }}_{2}\left(\theta \right)=L\left(1-e^{p_{1}\tau T}\right)e^{p_{2}\left(m-T\right)}-\frac{q}{p_{2}}.$$Then, the solution to (10) satisfying $\psi_{2}\left(\theta \right)={\tilde{\psi }}_{2}\left(\theta \right)=\psi_{hlc}$ is given by
(15)$$\begin{array}{c} {\tilde{\psi }}_{2}^{*}\left(t\right)=\left\{\begin{array}{c} \left(\psi_{hlc}+\frac{q}{p_{1}}\right)e^{p_{1}\left(t-\theta \right)}-\frac{q}{p_{1}},\;t\in \left[\theta ,\left(k_{1}+\tau \right)T\right],\\ \left(\psi_{hlc}+\frac{q}{p_{1}}\right)e^{p_{1}\left(t-kT-\theta \right)}-\frac{q}{p_{1}},\;t\in \left[\left(k_{1}+k\right)T,\left(k_{1}+k+\tau \right)T\right],\;k\in N_{+},\\ \left(\left(\psi_{hlc}+\frac{q}{p_{1}}\right)e^{p_{1}\left(\left(k_{1}+\tau \right)T-\theta \right)}-\frac{q}{p_{1}}+\frac{q}{p_{2}}\right)e^{p_{2}\left(t-\left(k_{1}+\tau \right)T-kT\right)}-\frac{q}{p_{2}},\\ \;t\in \left[\left(k_{1}+\tau +k\right)T,\left(k_{1}+1+k\right)T\right],k\in N_{0}, \end{array}\right. \end{array}$$
if $\theta \in [k_{1}T,k_{1}T+\tau T]$ and
(16)$${\tilde{\psi }}_{2}^{*}\left(t\right)=\left\{\begin{array}{c} \left(\psi_{hlc}+\frac{q}{p_{2}}\right)e^{p_{2}\left(t-\theta \right)}-\frac{q}{p_{2}},\;t\in \left[\theta ,\left(k_{1}+1\right)T\right],\\ \left(\psi_{hlc}+\frac{q}{p_{2}}\right)e^{p_{2}\left(t-kT-\theta \right)}-\frac{q}{p_{2}},\;t\in \left[\left(k_{1}+k+\tau \right)T,\left(k_{1}+k+1\right)T\right],\;k\in N_{+},\\ \left(\left(\psi_{hlc}+\frac{q}{p_{2}}\right)e^{p_{2}\left(\left(k_{1}+1\right)T-\theta \right)}-\frac{q}{p_{2}}+\frac{q}{p_{1}}\right)e^{p_{1}\left(t-\left(k_{1}+1\right)T-kT\right)}-\frac{q}{p_{1}},\\ \;t\in \left[\left(k_{1}+k\right)T,\left(k_{1}+\tau +k\right)T\right],k\in N_{0}, \end{array}\right.$$
if $\theta \in [k_{1}T+\tau T,\left(k_{1}+1\right)T]$. As observed, ${\tilde{\psi }}_{2}^{*}\left(t\right)={\tilde{\psi }}_{2}^{*}\left(t+kT\right)$ and ${\tilde{\psi }}_{2}^{*}\left(t\right)\in \left[min\left(-\frac{q}{p_{1}},-\frac{q}{p_{2}}\right),max\left(-\frac{q}{p_{1}},-\frac{q}{p_{2}}\right)\right]$; therefore, this solution is periodic and bounded.The boundedness of ${\tilde{\psi }}_{2}^{*}\left(t\right)$ and the state z(t) guarantees the fulfillment of the transversality condition
(17)$$\lim_{t\to +\infty }inf\;e^{\lambda (\theta -t)}{\tilde{\psi }}_{2}^{*}\left(t\right)\left(z\left(t\right)-z^{*}\left(t\right)\right)\geq 0,$$
for any admissible solution z(t).Thus, by considering the concavity of the Hamiltonian, we can conclude that $v^{*}$ is the optimal control, where
(18)$$v_{i}^{*}=\left\{\begin{array}{cc} b_{i}, & {\tilde{\psi }}_{2}^{*}\left(t\right)>0,\\ b_{i}+\frac{\xi_{i}}{a_{i}}{\tilde{\psi }}_{2}^{*}\left(t\right), & {\tilde{\psi }}_{2}^{*}\left(t\right)\in \left[-\frac{a_{i}b_{i}}{\xi_{i}},0\right],\\ 0, & {\tilde{\psi }}_{2}^{*}\left(t\right)<-\frac{a_{i}b_{i}}{\xi_{i}}. \end{array}\right.$$The uniqueness of the obtained optimal solution follows from the concavity of the Hamiltonian. Note that the Hamiltonian is a concave function with respect to the state *z* and strictly concave with respect to the control $v_{i}$, thus following the uniqueness of the obtained optimal solution. This completes the proof of the theorem. □

**Lemma 1.** 
*If ${\tilde{\psi }}_{2}\left(\theta \right)\neq \psi_{hlc}$, then $\lim_{t\to +\infty }∥{\tilde{\psi }}_{2}\left(t\right)∥=\infty$.*


**Proof.** When ${\tilde{\psi }}_{2}\left(\theta \right)$ deviates from the equilibrium initial value $\psi_{hlc}$, let ${\tilde{\psi }}_{2}\left(\theta \right)=\psi_{hlc}+c$. Then, from the moment θ, the difference of each period can be obtained:
$${\tilde{\psi }}_{2}\left(\theta +kT\right)-{\tilde{\psi }}_{2}\left(\theta +\left(k-1\right)T\right)=ce^{kT(p_{2}\left(1-\tau \right)+p_{1}\tau )}.$$For $k\in N_{+}$ and $kT(p_{2}\left(1-\tau \right)+p_{1}\tau )>0$, the value difference $ce^{kT(p_{2}\left(1-\tau \right)+p_{1}\tau )}$ varies monotonically. Thus, the value of ${\tilde{\psi }}_{2}$ diverges with time. This implies that when t→+∞, if c>0, ${\tilde{\psi }}_{2}\left(t\right)$ approaches +∞, and if c<0, ${\tilde{\psi }}_{2}\left(t\right)$ approaches −∞. □

**Lemma 2.** 
*Optimal control $v_{i}^{*}$ is environmentally sustainable when the following inequalities are satisfied:*

(19)
$$\left\{\begin{array}{cc} \left(\psi_{hlc}+\frac{q}{p_{1}}\right)e^{p_{1}\left(\left(k_{1}+\tau \right)T-\theta \right)}-\frac{q}{p_{1}}\geq -\frac{a_{i}b_{i}}{\xi_{i}}, & \theta \in [k_{1}T,\left(k_{1}+\tau \right)T],\\ \left(\left(\psi_{hlc}+\frac{q}{p_{2}}\right)e^{p_{2}\left(\left(k_{1}+1\right)T-\theta \right)}-\frac{q}{p_{2}}+\frac{q}{p_{1}}\right)e^{p_{1}\left(1-\tau \right)T}-\frac{q}{p_{1}}\geq -\frac{a_{i}b_{i}}{\xi_{i}}, & \theta \in [\left(k_{1}+\tau \right)T,\left(k_{1}+1\right)T], \end{array}\right.$$

*if $\delta_{1}>\delta_{2}$, $k_{1}=\left\lfloor \frac{\theta }{T}\right\rfloor$, and*

(20)
$$\left\{\begin{array}{cc} \left(\left(\psi_{hlc}+\frac{q}{p_{1}}\right)e^{p_{1}\left(\left(k_{1}+\tau \right)T-\theta \right)}-\frac{q}{p_{1}}+\frac{q}{p_{2}}\right)e^{p_{2}\left(1-\tau \right)T}-\frac{q}{p_{2}}\geq -\frac{a_{i}b_{i}}{\xi_{i}}, & \theta \in [k_{1}T,\left(k_{1}+\tau \right)T],\\ \left(\psi_{hlc}+\frac{q}{p_{2}}\right)e^{p_{2}\left(\left(k_{1}+1\right)T-\theta \right)}-\frac{q}{p_{2}}\geq -\frac{a_{i}b_{i}}{\xi_{i}}, & \theta \in [\left(k_{1}+\tau \right)T,\left(k_{1}+1\right)T], \end{array}\right.$$

*if $\delta_{1}<\delta_{2}$.*


**Proof.** For $\delta_{1}>\delta_{2}$, the dynamic of adjoint variable ${\tilde{\psi }}_{2}^{*}\left(t\right)$ decreases in the first subperiod and increases in the second subperiod. In addition, according to Theorem 1, ${\tilde{\psi }}_{2}^{*}\left(t\right)\;\in \;\left[min\left(-\frac{q}{p_{1}},-\frac{q}{p_{2}}\right),max\left(-\frac{q}{p_{1}},-\frac{q}{p_{2}}\right)\right]$. Then, we have ${\tilde{\psi }}_{2}^{*}\left(t\right)<max\left(-\frac{q}{p_{1}},-\frac{q}{p_{2}}\right)<0$.Hence, to ensure that optimal control $v_{i}^{*}$ is environmentally sustainable, the condition $\min_{t\geq 0}{\tilde{\psi }}_{2}^{*}\left(t\right)\geq -\frac{a_{i}b_{i}}{\xi_{i}}$ must be satisfied. As such, we have
(21)$$\left\{\begin{array}{cc} \min_{t\geq 0}{\tilde{\psi }}_{2}^{*}\left(t\right)={\tilde{\psi }}_{2}^{*}\left(\left(k_{1}+\tau \right)T\right)\geq -\frac{a_{i}b_{i}}{\xi_{i}}, & \theta \in [k_{1}T,\left(k_{1}+\tau \right)T],\\ \min_{t\geq 0}{\tilde{\psi }}_{2}^{*}\left(t\right)={\tilde{\psi }}_{2}^{*}\left(\left(k_{1}+\tau +1\right)T\right)\geq -\frac{a_{i}b_{i}}{\xi_{i}}, & \theta \in [\left(k_{1}+\tau \right)T,\left(k_{1}+1\right)T]. \end{array}\right.$$Similarly, the condition for $\delta_{1}<\delta_{2}$ can be obtained. □

Figure 1 illustrates the dynamics of ${\tilde{\psi }}_{2}$ with different initial values in two situations: when distribution shift parameter θ is located before and after the switching time in one period. Without loss of generality, period *T* and switching ratio τ can be assigned randomly because these two parameters do not affect the overall result. Hence, we denote T=1;τ=0.5. In addition, for parameter λ, which directly determines the expectation of the game-termination time, we denote λ=0.5. The other parameters are set as $\delta_{1}\;=\;0.9,\;\delta_{2}=0.45$, and q=6.

The blue line, the initial value ${\tilde{\psi }}_{2}\left(\theta \right)$ which is equal to the equilibrium value $\psi_{hlc}$, denotes an equilibrium solution of the adjoint variable, which varies periodically with equal amplitude within interval $\left[\psi_{22}^{*},\psi_{21}^{*}\right]$, where $\psi_{2i}^{*}=-\frac{q}{p_{i}},i=1,2$ are equilibrium positions for each mode, depending on the change of δ. The equilibrium positions are represented by the sky-blue dash lines, and the red lines indicate nonequilibrium solutions; their initial values slightly deviate from the equilibrium value. Thus, the application of a small deviation to the initial value can cause the solution to diverge over time, either going into +∞ or −∞. The solution can then escape from two equilibrium points $\left[\psi_{22}^{*},\psi_{21}^{*}\right]$.

In this way, an equilibrium solution is uniquely determined, forming an HLC as time approaches an infinite horizon with infinite time-driven switches.

### 3.2. First Interval—[0,θ]

The payoff functional of the first interval, [0,θ], is denoted as
(22)$$J_{co1}\left(z_{0},v_{1},v_{2},\ldots ,v_{n}\right)=\sum_{i=1}^{n}\int_{0}^{\theta }\left[a_{i}v_{i}\left(b_{i}-v_{i}/2\right)-q_{i}z\right]\;dt.$$

The Hamiltonian is given by
(23)$$H_{co1}\left(t\right)=\sum_{i=1}^{n}\left[a_{i}v_{i}\left(b_{i}-v_{i}/2\right)-q_{i}z\right]+\psi_{1}\left(\sum_{i=1}^{n}\xi_{i}v_{i}-\delta \left(t\right)z\right)$$

Then, we have the canonical system:(24)$$\begin{array}{cc} \; & \dot{z}=\sum_{i=1}^{n}\xi_{i}v_{i}-\delta \left(t\right)z,\\ \; & {\dot{\psi }}_{1}=\sum_{i=1}^{n}q_{i}+\delta \left(t\right)\psi_{1},\;\psi_{1}\left(\theta \right)={\tilde{\psi }}_{2}\left(\theta \right)=\psi_{hlc}. \end{array}$$

Moreover, the continuity condition $\psi_{1}\left(\theta \right)={\tilde{\psi }}_{2}\left(\theta \right)=\psi_{hlc}$ is based on the continuity of the optimal control, which is directly driven by the adjoint variable, based on the fact that the switching instants depend on time, i.e., are autonomous or time-driven [9,22].

According to the first order derivative of the Hamiltonian, the optimal control is obtained as follows:(25)$$v_{i}^{*}=\left\{\begin{array}{cc} b_{i}, & \psi_{1}>0,\\ b_{i}+\frac{\xi_{i}}{a_{i}}\psi_{1}, & \psi_{1}\in \left[-\frac{a_{i}b_{i}}{\xi_{i}},0\right],\\ 0, & \psi_{1}<-\frac{a_{i}b_{i}}{\xi_{i}}. \end{array}\right.$$

As the terminal value of $\psi_{1}\left(t\right)$ is determined, we consider the backward time dynamics of $\psi_{1}\left(t\right)$, which is described by the following system:(26)$${\dot{\psi }}_{1}\left(t\right)=\left\{\begin{array}{cc} -\sum_{i=1}^{n}q_{1}-\delta \left(t\right)\psi_{1}, & t\in [hT,(h+\tau )T),\\ -\sum_{i=1}^{n}q_{2}-\delta \left(t\right)\psi_{1}, & t\in [(h+\tau )T),(h+1)T)], \end{array}\right.$$
where $h\in N_{0}$, with $\psi_{1}\left(\theta \right)=\psi_{hlc}$. Each subsystem of (26) has a single stable equilibrium at $\psi_{1i}^{*}=-\frac{q_{i}}{\delta_{i}},\;i=1,2$.

Note the equilibrium points of adjoint variables in the first interval are less than those in the second interval; thus, the overall trend of adjoint variable $\psi_{1}\left(t\right)$ in the first interval, [0,θ], increases in the backward time.

Accordingly, back to the forward time point of view, the optimal control in the first interval, [0,θ], may first retain around the maximal admissible value and then decreases to the HLC.

### 3.3. Numerical Overall Adjoint Variable

The initial value of adjoint variable $\psi_{hlc}$ in the second interval is uniquely determined, and this is used as the terminal value in the first interval. Now, we can solve the cooperative adjoint variable, denoted by $\psi_{co}\left(t\right)$, in the whole time horizon [0,+∞]. Herein, the parameter settings were similar to those in Section 3.1.

Figure 2 shows the situation when distribution shift parameter θ is located before the switching time in the second period, T≤θ≤T+τ, where
(27)$$\psi_{co}\left(t\right)=\left\{\begin{array}{cc} \psi_{1}\left(t\right), & t\in [0,\theta ],\\ {\tilde{\psi }}_{2}\left(t\right), & t\in [\theta ,\infty ], \end{array}\right.\;\psi_{1}\left(\theta \right)={\tilde{\psi }}_{2}\left(\theta \right)=\psi_{hlc}.$$

Figure 3 shows the situation when distribution shift parameter θ is located after the switching time in the second period, T+τ≤θ≤2T.

The blue lines in the figure represent for the cooperative adjoint variable, $\psi_{co}\left(t\right)$; the sky-blue dash lines represent the equilibrium points of two intervals, and the green lines show the distribution shift parameter θ. From Figure 2 and Figure 3, we can conclude that regardless of whether shift instant θ is located before or after the switching time in a period, the overall trend of $\psi_{co}\left(t\right)$ does not change.

Consequently, the overall optimal control of player *i* in the whole time duration [0,∞] is derived as:(28)$$v_{i}^{*}=\left\{\begin{array}{cc} b_{i}, & \psi_{co}>0,\\ b_{i}+\frac{\xi_{i}}{a_{i}}\psi_{co}, & \psi_{co}\in \left[-\frac{a_{i}b_{i}}{\xi_{i}},0\right],\\ 0, & \psi_{co}<-\frac{a_{i}b_{i}}{\xi_{i}}. \end{array}\right.$$

## 4. Cooperative Game with Different Shifted Exponential Distributions

In reliability engineering, the pollution-control equipment used by each player is different and with a different warranty period. The duration of the warranty period may vary depending on factors such as product type, brand, and contract terms, etc., and it is usually measured in months or years. Hence, the *i*-th player’s equipment fails abruptly at moment $T_{fi}$ as a random variable with a known probability distribution function $F_{i}\left(t\right),\;i\;=\;\bar{1,n}$, and the equipment may break down owing to the end of its lifetime or other natural disasters. The game lasts until one of the players’ equipments breaks down. Hence, this study also considered an *n*-player cooperative game of pollution control with different shifted exponential distributions. Furthermore, each player is assumed to possess specific equipment used in pollution control. Moreover, ${\left\{T_{fi}\right\}}_{1}^{n}$ are assumed to be independent random variables. Thus, the game duration is defined as $T_{f}=min\left\{T_{f1},\ldots ,T_{fn}\right\}$.

In this case, players’ equipment is heterogeneous, and ${\left\{T_{fi}\right\}}_{1}^{n}$ adopts different shifted exponential distributions, as well as different distribution parameters, ${\left\{\lambda_{i}\right\}}_{1}^{n}$. Without loss of generality, we assumed that $\theta_{1}\leq \theta_{2}\ldots \leq \theta_{n}$, where $\theta_{n}$ is the shifted parameter of player *n*, and it represents the largest shifted parameter among all players. Then, we have $F\left(t\right)=P\left\{T_{f}<t\right\}=1-\prod_{i=1}^{n}\left(1-F_{i}\left(t\right)\right)$, see [15], where $F_{i}\left(t\right)$ is defined as
(29)$$F_{i}\left(t\right)=\left\{\begin{array}{cc} 1-e^{\lambda_{i}\left(\theta_{i}-t\right)}, & t>\theta_{i},\\ 0, & t\in [0,\theta_{i}]. \end{array}\right.$$

The duration of the game is a random variable with composite distribution function [23]. Thus, the cumulative distribution function, F(t), with different shifted exponential distributions is denoted as
(30)$$F\left(t\right)=\left\{\begin{array}{cc} 0, & t\in [0,\theta_{1}],\\ 1-e^{\lambda_{1}\left(\theta_{1}-t\right)}, & t\in [\theta_{1},\theta_{2}],\\ \ldots \\ 1-e^{\sum_{i=1}^{j}\left[\lambda_{i}\left(\theta_{i}-t\right)\right]}, & t\in [\theta_{j},\theta_{j+1}],\\ \ldots \\ 1-e^{\sum_{i=1}^{n}\left[\lambda_{i}\left(\theta_{i}-t\right)\right]}, & t>\theta_{n}. \end{array}\right.$$

The cooperative payoff functional, (4), in this case, can be rewritten as the following sum:(31)$$\begin{array}{cc} J_{co}(z_{0},v_{1},v_{2},\ldots , & v_{n})=J_{1}\left(z_{0},v_{1},v_{2},\ldots ,v_{n}\right)+J_{2}\left(z_{0},v_{1},v_{2},\ldots ,v_{n}\right)+\ldots +J_{n}\left(z_{0},v_{1},v_{2},\ldots ,v_{n}\right)\\ & =\sum_{i=1}^{n}[\int_{0}^{\theta_{1}}\left[a_{i}v_{i}\left(b_{i}-v_{i}/2\right)-q_{i}z\right]\;dt+\int_{\theta_{1}}^{\theta_{2}}e^{\lambda_{1}\left(\theta_{1}-t\right)}\left[a_{i}v_{i}\left(b_{i}-v_{i}/2\right)-q_{i}z\right]\;dt\\ & +\ldots +\int_{\theta_{n}}^{\infty }e^{\sum_{i=1}^{n}\left[\lambda_{i}\left(\theta_{i}-t\right)\right]}\left[a_{i}v_{i}\left(b_{i}-v_{i}/2\right)-q_{i}z\right]\;dt]. \end{array}$$

### 4.1. The Last Interval—$\left(\theta_{n},\infty \right]$

The payoff functional in the last interval, $\left(\theta_{n},\infty \right]$, is
(32)$$J_{co_{n}}\left(z_{0},v_{1},v_{2},\ldots ,v_{n}\right)=\sum_{i=1}^{n}\int_{\theta_{n}}^{\infty }e^{\sum_{i=1}^{n}\left[\lambda_{i}\left(\theta_{i}-t\right)\right]}\left[a_{i}v_{i}\left(b_{i}-v_{i}/2\right)-q_{i}z\right]\;dt.$$

The Hamiltonian is given by
(33)$$H_{co_{n}}\left(t\right)=e^{\sum_{i=1}^{n}\left[\lambda_{i}\left(\theta_{i}-t\right)\right]}\sum_{i=1}^{n}\left[a_{i}v_{i}\left(b_{i}-v_{i}/2\right)-q_{i}z\right]+\psi_{n}\left(\sum_{i=1}^{n}\xi_{i}v_{i}-\delta \left(t\right)z\right).$$

Then, we have the following canonical system:(34)$$\begin{array}{cc} \; & \dot{z}=\sum_{i=1}^{n}\xi_{i}v_{i}-\delta \left(t\right)z,\\ \; & {\dot{\psi }}_{n}=e^{\sum_{i=1}^{n}\left[\lambda_{i}\left(\theta_{i}-t\right)\right]}\sum_{i=1}^{n}q_{i}+\delta \left(t\right)\psi_{n}. \end{array}$$

Let ${\tilde{\psi }}_{n}=e^{-\sum_{i=1}^{n}\left[\lambda_{i}\left(\theta_{i}-t\right)\right]}\psi_{n}$; then, we have ${\dot{\tilde{\psi }}}_{n}=\sum_{i=1}^{n}q_{i}+\left(\delta \left(t\right)+\sum_{i=1}^{n}\lambda_{i}\right){\tilde{\psi }}_{n}$.

As the differential equation of ${\tilde{\psi }}_{n}\left(t\right)$ is similar to the identical shift case, ${\tilde{\psi }}_{2}\left(t\right)$, in the Section 2, we can still uniquely determine the equilibrium initial value, ${\tilde{\psi }}_{n}\left(\theta_{n}\right)\;=\;\psi_{hlc_{N}}$, that forms a unique HLC.
(35)$$\psi_{hlc_{N}}=\left\{\begin{array}{cc} \bar{L}\left(1-e^{p_{2N}T\left(\tau -1\right)}\right)e^{p_{1N}m_{n}}-\frac{q}{p_{1N}}, & m_{n}\in \left[0,\tau T\right],\\ \bar{L}\left(1-e^{p_{1N}\tau T}\right)e^{p_{2N}\left(m_{n}-T\right)}-\frac{q}{p_{2N}}, & m_{n}\in \left[\tau T,T\right], \end{array}\right.$$
where $q=\sum_{i=1}^{n}q_{i},\;\lambda_{N}=\sum_{i=1}^{n}\lambda_{i},\;p_{1N}=\delta_{1}+\lambda_{N},\;p_{2N}=\delta_{2}+\lambda_{N},\;m_{n}=\theta_{n}-\left\lfloor \frac{\theta_{n}}{T}\right\rfloor T,$ $\bar{L}=\frac{q(p_{1N}-p_{2N})}{p_{1N}p_{2N}\left(e^{p_{2N}T\left(\tau -1\right)}-e^{p_{1N}\tau T}\right)}.$

### 4.2. Former Intervals—$\left[0,\theta_{1}\right]\cup \left(\theta_{1},\theta_{2}\right]\cup \ldots \cup \left(\theta_{n-1},\theta_{n}\right]$

Next, the equilibrium value of the adjoint variable at moment $\theta_{n}$ is uniquely determined. Thus, the dynamics of $\psi_{i}\left(t\right),\;i=\bar{1,n-1}$ are also uniquely determined in backward time.

The payoff functional of player *j*, $j\in \left[1,n-1\right],\;j\in N_{0}$, is given by
(36)$$J_{co_{j}}\left(z_{0},v_{1},v_{2},\ldots ,v_{n}\right)=\sum_{i=1}^{n}\int_{\theta_{j}}^{\theta_{j+1}}e^{\sum_{i=1}^{j}\left[\lambda_{i}\left(\theta_{i}-t\right)\right]}\left[a_{i}v_{i}\left(b_{i}-v_{i}/2\right)-q_{i}z\right]\;dt.$$

Furthermore, the Hamiltonian is denoted as
(37)$$H_{co_{j}}\left(t\right)=e^{\sum_{i=1}^{j}\left[\lambda_{i}\left(\theta_{i}-t\right)\right]}\sum_{i=1}^{n}\left[a_{i}v_{i}\left(b_{i}-v_{i}/2\right)-q_{i}z\right]+\psi_{j}\left(\sum_{i=1}^{n}\xi_{i}v_{i}-\delta \left(t\right)z\right).$$

The differential equation of $\psi_{j}\left(t\right)$ is given as
(38)$${\dot{\psi }}_{j}=e^{\sum_{i=1}^{j}\left[\lambda_{i}\left(\theta_{i}-t\right)\right]}\sum_{i=1}^{n}q_{i}+\delta \left(t\right)\psi_{n}.$$

Let ${\tilde{\psi }}_{j}=e^{-\sum_{i=1}^{j}\left[\lambda_{i}\left(\theta_{i}-t\right)\right]}\psi_{j}$; then, we have
$${\dot{\tilde{\psi }}}_{j}=\sum_{i=1}^{n}q_{i}+\left(\delta \left(t\right)+\sum_{i=1}^{j}\lambda_{i}\right){\tilde{\psi }}_{j},\;\;{\tilde{\psi }}_{j}\left(\theta_{j+1}\right)={\tilde{\psi }}_{j+1}\left(\theta_{j+1}\right).$$

In addition, for the first interval $\left[0,\theta_{1}\right]$, we have ${\dot{\psi }}_{1}\left(t\right)=\sum_{i=1}^{n}q_{i}+\delta \left(t\right)\psi_{1}\left(t\right),$ $\psi_{1}\left(\theta_{1}\right)\;=\;\tilde{\psi_{2}}\left(\theta_{1}\right).$

Consequently, the overall cooperative adjoint variable with different shifted distributions is obtained as follows:(39)$$\psi_{co}\left(t\right)=\left\{\begin{array}{cc} \psi_{1}\left(t\right), & t\in [0,\theta_{1}],\\ {\tilde{\psi }}_{2}\left(t\right), & t\in [\theta_{1},\theta_{2}],\\ \ldots \\ {\tilde{\psi }}_{j}\left(t\right), & t\in [\theta_{j},\theta_{j+1}],\\ \ldots \\ {\tilde{\psi }}_{n}\left(t\right), & t>\theta_{n}. \end{array}\right.$$

## 5. Numerical Optimal Solution

For simplicity, we considered an example of a two-player cooperative game-theoretic model of pollution control. As the overall cooperative adjoint variable is uniquely obtained in each case, optimal solutions are shown below.

To meet the condition of environmentally sustainable control, the parameters settled were set as follows: $\delta_{1}=0.9;\delta_{2}=0.45;T=1;\;\tau =0.5;\lambda =0.5;q=6;\;\xi_{1}\;=\;0.8;$ $\xi_{2}\;=\;0.7;a_{1}=1;\;a_{2}=1.2;b_{1}=b_{2}=10;\;z_{0}=0;\;\theta =1.2;\theta_{1}=1;\;\theta_{2}=2.2;\;\lambda_{1}=0.3;$ and $\lambda_{2}=0.5$.

### 5.1. Solution of Identical Shifted Exponential Distribution

The optimal control, state trajectory, and corresponding cooperative payoff are demonstrated in Figure 4.

The solution shows that the optimal emissions of each player change periodically after the distribution shift instant θ=1.2. Further, the stock of pollution converged to a unique HLC and stabilized, and the cooperative payoff increased and then converged.

In some cases, before time instant θ, there may exist a period of radical emissions. This could be interpreted as a more intense use of equipment by players during the warranty period.

### 5.2. Solution of Different Shifted Exponential Distributions

The optimal control, state trajectory, and corresponding cooperative payoff in this case are shown in Figure 5.

Figure 5 shows that after the expiration of each piece of equipment’s warranty period ($\theta_{i}$), the control strategy of player *i* tends to be increasingly conservative. After the maximal warranty period $\theta_{2}$, the control strategy of each player transforms to a periodic solution in the form of an HLC. The stock of pollution also converges to a unique HLC and stabilizes, while the cooperative payoff increases and then converges. This result indicates that the proposed control strategy can still maintain profitability when the equipment of each player is nonhomogeneous with seasonal fluctuation (δ(t)).

For the dynamic switched system of pollution levels of a lake considered in this paper, based on the obtained results and coupled with the random duration of the process, we proposed that, within the framework of a cooperative game, players may adopt a production strategy transitioning from aggressive to conservative (gradually decreasing output) before the maximum warranty period $\theta_{2}$. This strategy reaches its equilibrium value at $\theta_{2}$. Moreover, after $\theta_{2}$, players adopt a production strategy following an HLC pattern. This involves the gradual increase and progressive decrease in production during periods of relatively high and low lake self-cleaning rates, respectively. Consequently, this approach ensures the attainment of an optimally controlled outcome from the context of sustainable development.

Therefore, based on the considerations of real-world issues, the obtained results exhibit uniqueness, theoretical applicability, and practical relevance.

## 6. Conclusions

In this study, we analyzed the cooperative differential game for a typical hybrid optimal pollution-control problem with two types of time-driven switches: the seasonal fluctuation(self-cleaning rate of the lake) and the shifted parameter of exponential distributions due to the random game duration. A random terminal duration problem was transformed into a combination of an infinite horizon and a finite horizon(s) optimal control problem.

Further, we first considered a scenario with identical game duration and then examined another scenario with joint probability distribution of game duration resulting from the heterogeneity of players’ equipment. This paper discussed these two scenarios in detail and presented the results of each scenario analytically and numerically. Furthermore, an environmentally sustainable solution in the form of an HLC was uniquely determined for each scenario, ensuring both sustainable production revenue and environmental protection.

In a subsequent study, we will delve into the optimal control problem of each player in a noncooperative game featuring infinite time-driven switches and random game duration. In addition, we will provide comparisons of the results obtained from the noncooperative game with those obtained from the cooperative game to establish a justifiable allocation rule.

## Figures and Tables

**Figure 1 entropy-25-01426-f001:**
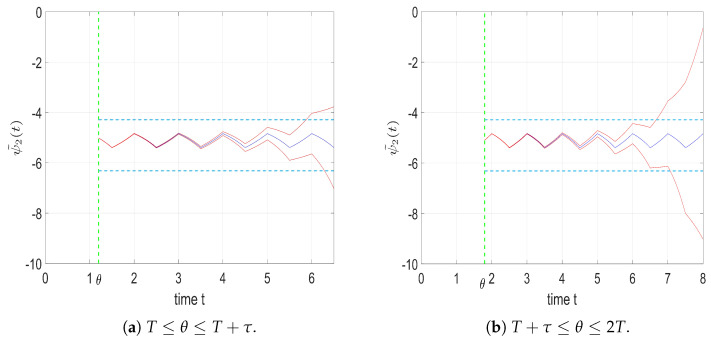
Dynamics of the ${\tilde{\psi }}_{2}$ with different initial values.

**Figure 2 entropy-25-01426-f002:**
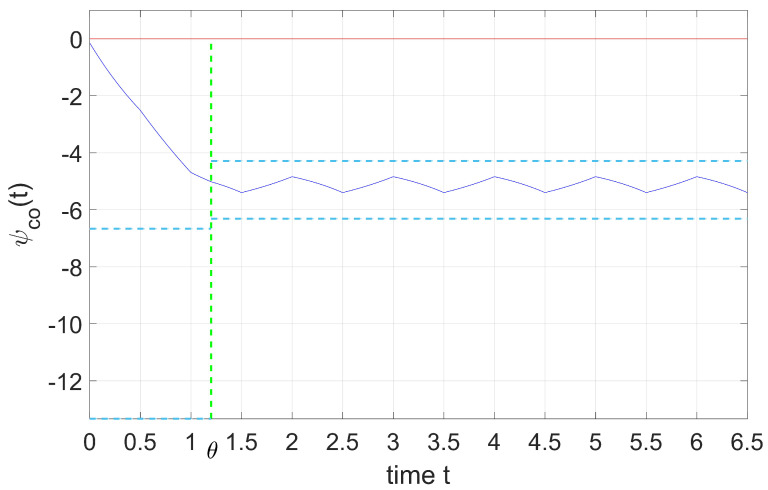
Dynamics of the $\psi_{co}$ if kT≤θ≤(k+τ)T.

**Figure 3 entropy-25-01426-f003:**
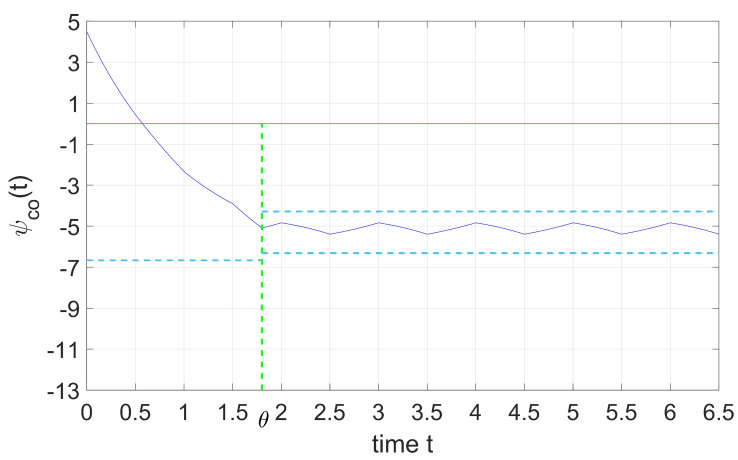
Dynamics of the $\psi_{co}$ if kT+τ≤θ≤(k+1)T.

**Figure 4 entropy-25-01426-f004:**
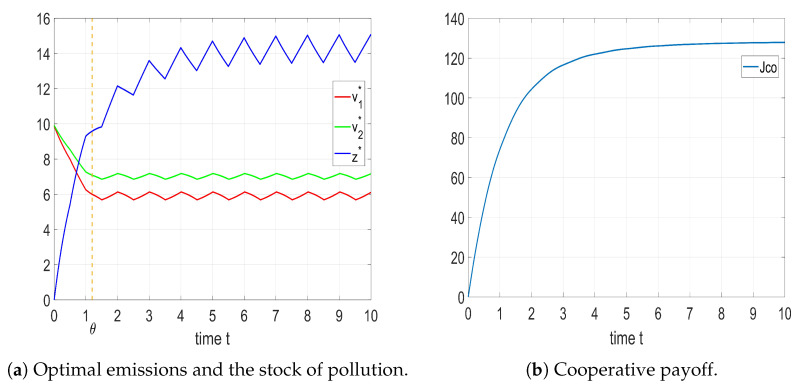
Optimal solution and cooperative payoff with identical shifted exponential distribution.

**Figure 5 entropy-25-01426-f005:**
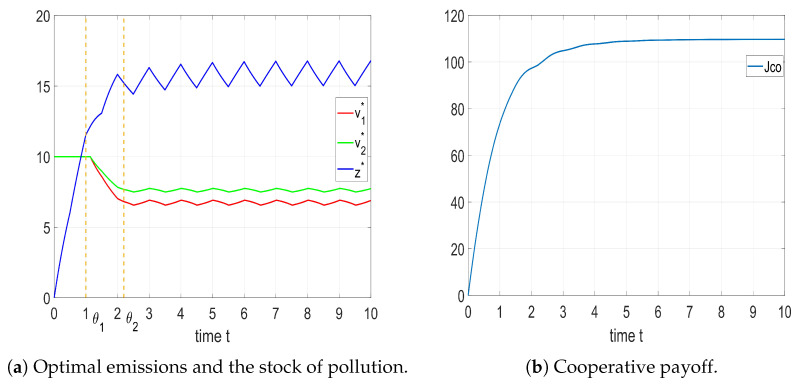
Optimal solution and cooperative payoff with different shifted exponential distributions.

## Data Availability

Not applicable.

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
