# Peer review of "Sustainable Optimal Control for Switched Pollution-Control Problem with Random Duration"

_entropy, 2023, doi:10.3390/e25101426_

Round 1

Reviewer 1 Report

The subject of interest is an application of differential games to the pollution-control problem with random duration. Formally, this is a multi-person game. The cooperative nature of the game makes the issue less complex. The level of difficulty is similar to that of the conventional one-agent decision-making procedure. Moreover, the utility function has a special quadratic form, and the differential equation describing a pollution stock has a linear formula. The mathematical model is too simple, which is why this paper does not convince me in both mathematical and theoretical terms. 

Reviewer 3 Report

This paper deals with a sustainable optimal control for switched pollution–control problem with random duration. This paper is well organized and well prove that to solve gibe problem. However, I would like to point out following as.

1.     In introduction, the description of previous work needs to their advantage and disadvantage, and also the description of object is ambiguous even though authors mentioned about it.

2.     Equation (2) is correct?

3.     In order to easily understand by reader, I hope authors should add entire figure or diagram.

4.     Please add recently published papers.

Please check the grammar.

Round 2

Reviewer 1 Report

The model is simple, and the result is not general. I admit that the general formulation of differential games is difficult. There is no need to make generalizations regarding the outcomes if the expense is excessively high. I found this paper to be an important contribution to cooperative game theory for sustainable development. For this reason, I find this paper to be publishable.

Reviewer 3 Report

I think this paper is well revised according to reviewer's point out. Thus, I would like to decided as an "accept"

Nothing